# Synthesis and Biological Activities of Cyclodepsipeptides of Aurilide Family from Marine Origin

**DOI:** 10.3390/md19020055

**Published:** 2021-01-23

**Authors:** Synthia Michon, Florine Cavelier, Xavier J. Salom-Roig

**Affiliations:** Institut des Biomolécules Max Mousseron (IBMM), UMR 5247, Université de Montpellier, CNRS, ENSCM, Place Eugène Bataillon, 34095 Montpellier, France; Synthia.Michon@etu.umontpellier.fr

**Keywords:** marine drugs, aurilides, depsipeptides, total synthesis, macrocyclization

## Abstract

Aurilides are a class of depsipeptides occurring mainly in marine cyanobacteria. Members of the aurilide family have shown to exhibit strong cytotoxicity against various cancer cell lines. These compounds bear a pentapeptide, a polyketide, and an α-hydroxy ester subunit in their structure. A large number of remarkable studies on aurilides have emerged since 1996. This comprehensive account summarizes the biological activities and total syntheses of natural compounds of the aurilide family as well as their synthetic analogues.

## 1. Introduction

Since the late 1960s, marine organisms have been intensively explored as hopeful resources for anticancer drugs, and a diverse array of new compounds are still being discovered every year. In particular, the sea hare *Dolabella auricularia* is known as a prolific producer of cytotoxic and/or antitumor structurally unique secondary metabolites such as dolastatins 10 and 15 [1]. From a specimen of this marine organism collected in the Japanese sea, Suenaga et al. [2] isolated in 1996 aurilide (**1**), a 26-membered cyclodepsipeptide, which exhibits a strong cytotoxicity against HeLa S_3_ cells with an IC_50_ of 0.011 μg/mL.

In the next two decades, the aurilide family expanded with the discovery of ten new members, which were structurally analogous to aurilide (structures of this family are shown in Figure 1).

As shown in Table 1, cyanobacterium *Lyngbya majuscule* was the source of aurilide B (**2**) and C (**3**), and cephalaspidean mollusc *Philinopsis speciosa* furnished kulokekahilide-2 (**4**). Palau’amide (**5**), odoamide (**6**), and lagunamides A (**7**), B (**8**), C (**9**), D (**10**), and D’ (**11**) were isolated from different sources of marine cyanobacteria.

Since the first synthesis of aurilide in 1996 by Suenaga and co-workers [2], the aurilide family has attracted considerable attention from the synthetic community due to potent antiproliferative activity as well as synthetically challenging molecular architecture and additional total syntheses of aurilide and other members of the aurilide class have been reported.

This review is intending to provide an overview of the synthesis and biological activities of these fascinating natural products. After focusing on the structural features and comparison of the biological activities of the aurilide class members, this report reviews the synthetic studies of the natural products and their analogues.

## 2. Structural Features

Structurally, the aurilide family members can be described as cyclic depsipeptides, whose framework can be divided into three subunits: an α-hydroxy acid residue, a polyketide segment containing three or four stereogenic centers, and a pentapeptide.

The size of the macrocycle differs slightly; eight of them (**1**−**4**, **6**−**8** and **10**) are 26-membered rings, and lagunamide C (**9**) is a 27-membered ring, while plau’amide (**5**) and lagunamide D’ (**11**) are 24-membered rings.

### 2.1. Differences in the α-Hydroxy Acid and the Polyketide Subunits

Three different α-hydroxy acids (Figure 2) can be found in the macrocyclic structure of the aurilide family members. 2-Hydroxyisoleucic acid (Hila) is the most common residue and can be found in aurilide (**1**), aurilide B (**2**), lagunamides A (**7**), B (**8**), C (**9**), D (**10**), and D’ (**11**) and odoamide (**6**). Kulokekahilide-2 (**4**) and palau’amide (**5**) bear a 2-hydroxyisocaproic acid (Hica), whereas 2-hydroxyisovaleric acid (Hiva) is exclusive to aurilide C (**3**).

The polyketide subunit consists of an α,β-unsaturated 5,7-dihydroxy acid bearing three contiguous stereocenters at C5, C6, and C7 with the relative configuration 5,6- and 6,7- *anti* and an aliphatic saturated or unsaturated side-chain at C8, whose nature is depending on the member of the family (Figure 3). Odoamide (**6**) and lagunamide A (**7**) bear a fourth stereocenter in the side chain, whose absolute configuration is *R* for the first compound and *S* for the second one. Lagunamide C (**9**) can be considered as a subclass of aurilide-related compounds that has a ring expansion due to the additional methylene carbon inserted in the polyketide-derived moiety at C7.

### 2.2. Differences in the Pentapeptide Fragment

Most of the depsipeptides of the aurilide class share similar peptide substructures. However, some of them have unique structural motifs and particularly in aurilides, three of the amino acids have different side chains. The first amino acid (AA_1_, Table 2) is l-*N*-methylalanine with the sole exception of kulokekalide-2 (**4**) in whose structure the other enantiomer is present. The third amino acid (AA_3_) is in all cases sarcosine, and the fifth one (AA_5_) is l-alanine, except for aurilides (**1**−**3**), which bear an l-valine. The main differences concern the second and fourth amino acid. Concerning the second amino acid (AA_2_), l-valine is exclusive to aurilides (**1**−**3**), while l-isoleucine is present in the other members of the family, although the absolute configuration of the side-chain stereocenter is not always the same (usually *S* but *R* in two cases). The fourth amino acid (AA_4_) normally found is d-phenylalanine, but in the case of aurilide (**1**), AA_4_ is replaced by l-leucine and in aurilides B (**2**) and C (**3**), it is replaced by d-isoleucine.

## 3. Biological Activities

Different members of the aurilide family exhibit strong cytotoxicity against various cancer cell lines, often in the range of nM concentrations, as summarized in Table 3.

Aurilide has been identified as an inducer of apoptosis by interfering with the morphogenesis of mitochondria [12]. Investigation of the mechanism of cytotoxicity showed that aurilide (**1**) binds to prohibitin 1 (PHB1), a mitochondria inner membrane protein, which in turn activates the proteolytic processing of optic atrophy 1 (OPA1), leading to mitochondrial fragmentation and apoptosis [13].

Lagunamide A induces caspase-mediated mitochondrial apoptosis in A549 cells [14]. Indeed, lagunamide A causes mitochondrial dysfunction followed by cell death along with the dissipation of mitochondrial membrane potential (Δφm) and overproduction of reactive oxygen species (ROS). It was proved that both anti- and pro-apoptotic B-cell lymphoma 2 (Bcl-2) family proteins, especially myeloid cell leukemia-1 (Mcl-1), participated in lagunamide A-induced mitochondrial apoptosis. The overexpression of Mcl-1 partly rescued A549 cells from lagunamide A-induced apoptosis.

Moreover, lagunamides A (**7**), B (**8**), and C (**9**) displayed significant antimalarial properties, with IC_50_ values of 0.19, 0.91, and 0.29 *μ*M, respectively, when tested against *Plasmodium falciparum* [7].

## 4. Structural Determination and Synthesis of the Natural Members of the Aurilide Family

### 4.1. Aurilides

After the isolation and structure elucidation of aurilide (**1**), it was important to confirm the absolute stereochemistry of the three contiguous stereogenic centers in the polyketide moiety (dihydroxy acid) [2,15]. For this purpose, **1** was treated with LiAlH_4_ followed by acylation with *p*-BrC_6_H_4_COCl in the presence of pyridine, leading to tris(*p*-bromobenzoate) **12d** (Scheme 1). Four stereoisomers of **12d** were prepared in order to compare them with the one obtained from the natural product.

Their synthesis began with an aldol reaction between the *Z* boron enolate of oxazolidinone **13** and *trans*-2-methyl-2-pentenal that gave the Evans/1,2-*syn* aldol adduct **14** (80% yield, dr > 95:5), which led to aldehyde **15** in three steps. The reaction between lithium anion of *tert*-butylacetate and aldehyde **15** provided a diastereomeric mixture of diols **16a** and **16b**, which were further transformed into conjugated esters **17a** and **17b** in four steps. Tris(*p*-bromobenzoates) **12a** and **12b** could be respectively accessed from **17a** and **17b** by a three-step sequence of reactions. In contrast, the synthesis of **12c** and **12d** required the transformation of **17a** and **17b** into enones **18a** and **18b**. Stereoselective ketone reduction followed by alcohol deprotection, ester reduction, and acylation of the corresponding triol afforded the expected tris(*p*-bromobenzoates) **12c** and **12d**. The ^1^H NMR and CD spectroscopic data for **12d** were entirely consistent with those obtained from the molecule generated from natural aurilide (**1**), thus establishing the absolute stereochemistry.

Following their initial work, the same authors described in 1997 the first total synthesis of aurilide (**1**) [15,16]. As shown in Scheme 2, the pentapeptide subunit **19** was accessed starting from sarcosine *tert*-butyl ester hydrochloride in four steps in 84% overall yield. The elongation on the *N*-terminal side uses the protecting group Z, which is quite unusual in such repetitive synthesis, and the last amino acid is coupled on the C-terminal side, which is possible, since sarcosine is not chiral.

The synthesis of polyketide moiety **20** is depicted in Scheme 3. For the introduction of the C-7 stereocenter, an alternative strategy involving an *anti*-Evans aldolisation was devised. Propionyl-2-oxazolidinone *ent*-**13** underwent a highly diastereoselective boron-mediated aldol reaction with *trans*-2-methyl-2-pentenal in ether to afford the *anti*-aldol **21** as the sole stereoisomer in 67% yield. An amide transformation of **21** with *N*,*O*-dimethylhydroxylamine, alcohol protection as *tert*-butyldimethylsilyl (TBS) ether, and diisobutyl aluminium hydride (DiBAL-H) reduction provided aldehyde **22** in 78% yield from **21**. Mukaiyama aldol reaction of **22** with 2-methyl-1-trimethylsiloxy-1,3-butadiene, followed by the oxidation of the corresponding aldehyde and treatment with diazomethane delivered the methyl ester **23**, which required the C5 stereochemistry inversion. Thus, after the oxidation of the free alcohol using Dess–Martin protocol, ketone **24** was diastereoselectively reduced with NaBH_4_ to afford the monoprotected diol **25** in 81% yield. The synthesis was completed by the protection of the hydroxyl group as an methylthiomethyl (MTM) ether followed by the ester hydrolysis to afford the protected dihydroxy acid **20** in 68% yield after two steps.

Having the building blocks **19** and **20** in hand, the attention was focused on their coupling using EDCI∙HCl and 4-(dimethylamino)pyridine (DMAP) to give the corresponding ester **26**. *tert*-Butylsilylether cleavage, esterification of the resulting free alcohol **27** with Fmoc-*N*-methyl-l-alanine, and removal of trichloroethyl group delivered **28**. Fmoc group cleavage and macrolactamization of the linear precursor by reaction with Bop-Cl and methylthiomethyl ether cleavage ultimately provided aurilide (**1**). This convergent strategy has the drawback that the C5 stereochemistry must be reversed by an oxidation/reduction sequence.

The synthesis of the first analogues of aurilide (**1**) has been disclosed by Takahashi et al. [17]. They prepared the natural product and a library of 25 analogues on solid support using Multipin methodology with the final macrocyclization being performed in solution. The supported tetrapeptides used to create the library were efficiently synthesized with the following eight amino acids: l and d-valine, l and d-methylleucine, l and d-leucine, glycine and sarcosine in an *N*-Fmoc protected version (Scheme 4).

For the preparation of the polyketide moiety, they followed the strategy reported by Suenaga (Scheme 5) [15,16]. Thus, an Evans aldol reaction between the *N*-acyloxazolidinone *ent*-**13** and (*E*)-2-methyl-2-pentenal in dry ether at −100 °C delivered a 1:1 mixture of two diastereomeric aldol products **21** and **29**, which could be separated by column chromatography on silica gel, but this considerably reduces the synthetic interest of this approach. Compound **21** was further elaborated to provide the monoprotected dihydroxy ester **30** [15,16]. Compound **30**, after protection of the free alcohol as an MTM ether and ester hydrolysis with LiOH, was coupled to d-*allo*-isoleucic acid ester **33** and Fmoc-*N*-methyl-L-alanine, with the previous removal of the TBS group, yielding **31**. Deprotection of phenacyl ester delivered **32**. The 7*R* stereoisomer **34** was prepared from **29** following the same strategy.

The assembly of the aliphatic acid **32** and the solid-supported tetrapeptide **35** for the completion of the synthesis of aurilide (**1**) was performed using the DIC/HOBt method (Scheme 6). Then, the Fmoc protected amine was unmasked and, after cleavage from the solid support, the linear precursor **36** was obtained. The subsequent macrocyclization was performed in the presence of 1-ethyl-3-[3-(dimethylamino)propyl]carbodiimide (EDCI) and 1-hydroxy-7-azabenzotriazole (HOAt), and the MTM group was cleaved using silver nitrate, affording aurilide (**1**) in 11% overall yield and a low purity of 49%.

Next, a library of aurilide analogues was prepared (Scheme 7). For this purpose, polyketide **34**, instead of **32** corresponding to the natural aurilide, was chosen because of its easier synthetic access. Various sequences of four amino acids involving l and d-Val, l and *N*-Me-d-Leu, l and d-Leu, sarcosine and glycine were synthesized (**37**) (Table 4) and coupled to the aliphatic moiety **34**. The reagents and conditions were the same as the ones outlined in Scheme 6 for the preparation of aurilide (**1**). In our opinion, the low purity of the tetrapeptides and particularly, the choice of the non-natural polyketide fragment **34** are the major drawbacks of this strategy. So far, the cytotoxicity of these aurilide analogues has not been reported.

Takahashi and co-workers [17] reported the synthesis of eight non-natural analogues of aurilide (**1**) and studied the structure–activity relationships. The effect of the C5 hydroxyl function was first investigated by preparing deoxyaurilide **40** (Scheme 8). The dehydration of **41**, previously prepared according to the strategy reported by Suenaga [15], was carried out by reaction with *o*-NO_2_C_6_H_4_SeCN and PBu_3_. The corresponding α,β,γ,δ-unsaturated ester was selectively hydrogenated, leading to an α,β-unsaturated ester, which after saponification using LiOH was converted into carboxylic acid **42**. By the condensation of **42** with **43** and TBS group removal, the hydroxy ester **44** was obtained. Esterification with Fmoc-*N*-Me-alanine, deprotection of Fmoc and MTM groups, and macrolactonization afforded deoxyaurilide **40**.

Then, they introduced an amine in the aurilide structure with the objective of using this function to conjugate a functional molecule such as biotin to search a target molecule. For this purpose, they prepared analogues **46**−**51**. Thus, **46** was accessed from derivative **52** [15] by esterification with protected lysine followed by macrolactamization and cleavage of the MTM group (Scheme 9). For the synthesis of analogue **47**, tetrapeptide **53** [15] was used as a substrate. The *t*Bu ester group was cleaved, and the resulting carboxylic acid was condensed with protected Lys providing pentapeptide **54**. After esterification of **54** with **55**, a similar sequence of reactions as described in Scheme 8 delivered the aurilide analogue **47**.

Synthesis of analogue 48 (Scheme 10) began with cleavage of the *Z* group of **56** followed by coupling with protected Lys giving tripeptide **57**. Further coupling with isoleucic acid at the *N*-terminus and with Val at the C-terminus resulted in the pentapeptide **58**, which was then esterified with carboxylic acid **59**. Further TBS group deprotection, coupling with Fmoc-*N*-Me-l-Ala and four-step sequence of reactions, including macrolactonization afforded analogue **48**.

Analogue **49**, which was accessed by acylation of aurilide (**1**), was the precursor of the new analogue **50**. Finally, analogue **51** was obtained from 6-*epi*-aurilide (**45**) as described above.

Table 5 summarizes the cytotoxicity of aurilide and analogues against HeLaS_3_ cells. The results show that deoxyaurilide **40** is slightly less cytotoxic than aurilide (**1**); therefore, the hydroxyl group of the natural product is not essential for its high cytotoxicity. Analogues **46**−**50** displayed considerable activity and consequently could be useful as a probe molecule to search for the target responsible for the activity.

In 2006, Han and co-workers [3] detailed the discovery of two natural analogues of aurilide, which they called aurilide B (**2**) and aurilide C (**3**). They established the planar structures and the absolute configurations of these natural products by chiral HPLC, NMR, and high resolution fast atom bombardment mass spectrometry (HR-FABMS) studies. These compounds have never been synthesized.

### 4.2. Kulokekahilide-2 *(**4**)*

Kulokekahilide-2 was isolated in 2004 by Nakao and co-workers [4], who elucidated its structure by spectroscopic analysis and chemical degradation. In order to confirm the results of their studies, they implemented diastereoselective synthesis of triols **59a**, **59b**, **59c**, and **59d** by using the strategy reported by Suenaga [15] (Scheme 11). The synthesis of **59a** and **59c** involved an Evans *syn*-aldol reaction between the acylated chiral oxazolidines **13** and **60** and (*E*)-2-methylbut-2-enal yielding **61a** and **61c** as precursors of the unsaturated esters **62a** and **62c**, which were converted into the desired triols **59a** and **59c** in three and two steps, respectively. Units **59b** and **59d** were accessed by using the protocol described by Heathcock, which furnished the *anti*-aldol products **61b** and **61d** as precursors of **62b** and **62d**. The latter were transformed into triols **59b** and **59d** in one and four steps, respectively.

The authors correlated **59d** with the triol issued from degradation of the natural kulokekahilide-2. The spectroscopic studies led to the conclusion that the structure of this natural product was **63** (Figure 4). In 2007, the same team synthesized the supposed kulokekahilide-2 **63** and accidentally the 2′-epimer of **63** [18].

For this purpose, as depicted in Scheme 12, they prepared the aliphatic acid **64** from monoprotected dihydoxy ester **65** following the methodology described in Scheme 5. The pentapeptide domain **66** was accessed from l-Ala-OTce∙HCl in eight steps, and hydroxy acid **64** was coupled to the resulting pentapeptide yielding **66**. After TBS and trichloroethyl (Tce) groups cleavage, the seco acid **67** was obtained. A macrolactonization followed by MTM group removal yielded 2′-*epi*-**63**. The configuration of l-Ala was completely reversed during the macrolactonization step. In a second attempt, the synthesis of **63** could be finally achieved from **68**.

Since both compounds **63** and 2′-*epi*-**63** were tested and none of them showed the expected cytotoxicity observed with the natural product (Table 6), the authors concluded that the structure of kulokekahilide-2 should be re-examined.

In this context, shortly afterwards, in order to confirm the structure of kulokekahilide-2 (**4**), four other stereoisomers (**69**–**72**) were prepared by the same group using a similar strategy. The ^1^H NMR spectrum of **69** was clearly consistent with the one of the natural product and showed potent cytotoxicity against two cell lines, providing strong evidence that the absolute stereochemistry of kulokekahilide-2 (**4**) involved the combination 14′-l-Ala, 11′-d-MePh, and 2′-d-Ala (Table 6) [19,20].

Shortly after [21,22], the same authors undertook the preparation of a series of new 26-membered analogues of kulokekahilide-2 (**73**–**82**) and determined their biological activities (Table 6). These studies provided new findings such that the *S* configuration in C14′ was essential to preserve cytotoxicity, while the configurations in C11′- and C2′ have only a slight influence. Protection of the C5 hydroxyl function had no influence on the activities. Interestingly, adding a chlorine atom in the para position of the phenyl group in the d-MePhe amino acid increases cytotoxicity. Finally, by comparing **69** (Kulokekahilide-2 (**4**)) and its analogues with aurilide (**1**), it seems that stereochemistry is a more important factor for biological activity than the nature of the different substituents in α-hydroxy acid and pentapeptide fragments.

In 2009, Umehara and co-workers [23] discovered that an intramolecular ester exchange occurred between C5 and C7 hydroxyl functions of the polyketide in the 26-membered kulokekahilide-2 (**4**), resulting in a 24-membered isomer **83** (Table 6). Both isomers were shown to be in equilibrium and displayed similar cytotoxicity. They synthesized three new 24-membered analogues (**83**–**85**) and determined their biological activities (Table 7) [21].

More recently, Han and co-workers [24] have considered performing positron emission tomography (PET) studies with kulokekahilide-2 (**4**). To do this, they synthesized a partial structure of an ^11^C-labeled C1-C10 partial structure of this member of the aurilide family: the monoprotected dihydroxy ester **87**.

As depicted in Scheme 13, they first prepared the nonradioactive analogue **88** from (*E*)-2-methylbut-2-enal and chiral oxazolidinone *ent***-60** via monoprotected diol **89** by following the procedure they previously described with minor modifications [4]. The synthesis of the radioactive derivative **87** was carried out according to a similar synthetic route starting from tiglic aldehyde and *ent*-**60**. Thus, the *p*-methoxybenzyl (PMB) monoprotected dihydroxy ester **90** was subjected to a cross-coupling metathesis with pinacol alkenylboronate **91**, using Grubbs II catalyst, to give **92** in poor yield. Adduct **92** was finally converted into the final target molecule **87** in 72% yield by a Pd^0^-mediated C-[^11^C]methylation using [^11^C]methyl iodide and Pd_2_(dba)_3_ in the presence of P(*o*-tolyl)_3_ and K_2_CO_3_ as a base.

### 4.3. Palau’amide *(**5**)*

In 2003, the discovery and structure elucidation of palau’amide (**5**), a new member of the aurilide family, was reported by Williams and co-workers [5].

The first total synthesis was carried out two years later by Zou et al. (Scheme 14) [25]. In order to achieve control of the absolute configurations of stereocenters C6 and C7 in the 5,7-dihydroxy-2,6-dimethyldodec-2-en-11-ynoic acid unit **93**, they used Oppolzer’s *syn*-aldolisation methodology involving the reaction of *N*-propionylcamphorsultam **94** with Et_2_BOTf, using DIPEA as a base, which was followed by the addition of 5-hexynal, leading to the *syn*-aldol product **95**. Further elaboration of **95** by reductive removal of the chiral auxiliary led to the corresponding primary alcohol, which was masked as its *tert*-butyldimethylether affording **96**. Then, the absolute configuration of the free secondary alcohol was reversed via a Mitsonobu-type protocol, the newly generated alcohol was protected as TBS-ether, and primary alcohol was oxidized into the corresponding aldehyde after removal of the TBS protecting group. Having aldehyde **97** in hand, a vinylogous Mukaiyama aldol reaction was implemented to generate the last stereogenic center of the polyketide subunit by reaction with (*E*)-(2-methylbuta-1,3-dienyloxy)-trimethylsilane **98** in the presence of boron trifluoride diethyl etherate affording **99** in 65% yield. The aldehyde function in **99** was oxidized, and the resulting carboxylic acid coupled with d-leucine-derived alcohol **100**. Then, an oxidation/reduction strategy was implemented to reverse the C5 absolute configuration. Thus, the free secondary alcohol underwent a Dess–Martin oxidation, and the subsequent diastereoselective reduction of the resulting ketone with NaBH_4_ yielded compound **93**.

Then, their attention was turned to the coupling of building block **93** with the *N*-Fmoc protected dipeptide **101**, which afforded **102** (Scheme 15). Removal of the alloc group and coupling with the previously Fmoc deprotected dipeptide **103** gave rise to **104**. Cleavage of allyl ester and Fmoc protecting groups was followed by a macrolactonization and a TBS deprotection delivering palau’amide (**5**). This convergent route has as original feature the development of an alternative approach to the synthetic route reported by Suenaga [15] for the synthesis of the polypeptide subunit. However, this approach suffers from the drawback that the C5 stereochemistry requires a two-step inversion.

Another alternative approach for the synthesis of the polyketide subunit of palau’amide (**5**) was proposed by Mohapatra and Nayak (Scheme 16) [26]. Their synthesis of C33–C44 fragment **105** commenced by the regioselective ring opening of epoxide **106** with Me_2_CuCNLi_2_, which afforded the desired monoprotected triol **107** along with the corresponding 1,2-diol in a ratio of 8:1. The mixture was exposed to NaIO_4_ in order to eliminate the minor regioisomer. Next, the three-step sequence reactions consisted of the chemoselective primary hydroxyl group benzoylation by reaction with BzCl, TBS ether formation of the secondary alcohol with TBSOTf, and benzoyl cleavage by treatment with K_2_CO_3_ in MeOH to yield alcohol **108**. Swern oxidation of the primary alcohol was conducted to the corresponding aldehyde, which was reacted in situ with trimethylsilyl (TMS) protected pentynylmagnesium bromide, yielding alcohol **109** as a 1:1.5 mixture of two stereoisomers. Ensuing TMS cleavage, oxidation of the corresponding mixture of alcohols and stereoselective reduction of the resulting ketone function with NaBH_4_ in the presence of CeCl_3_ gave the corresponding *syn*-1,3-diol. Protection of the newly generated hydroxyl group as a methoxymethyl (MOM) ether and PMB cleavage gave **110**. Oxidation of the primary alcohol with 2-iodoxybenzoic acid (IBX) resulted in the formation of the corresponding aldehyde, which was reacted with Wittig reagent [(allyloxycarbonyl)ethylene]triphenylphosphorane to give exclusively the *E* stereoisomer of the corresponding alkene. Finally, the deprotection of TBS ether using tetra-*n*-butylammonium fluoride (TBAF) gave rise to the polyketide segment **105**. The low stereoselectivity observed in the addition of the pentynylmagnesium bromide constitutes a current limitation of this approach.

The synthesis of four diastereomers of palau’amide (**5**) was achieved by Sugiyama et al. [27] combining the different possible absolute configurations at C5 and C6 in the polyketide moiety. Initial investigations were directed toward the preparation of the polyketide units **111a**, **111b**, **111c**, and **111d** (Scheme 17). For the synthesis of **111a** and **111b**, the key step of this new strategy was the asymmetric crotylboration involving the chiral ester boronate **112**, which was reacted with aldehyde **113** delivering quantitatively homoallylic alcohol **114** as a single diastereomer. The newly formed hydroxyl group was protected as a *tert*-butylsilyl ether, the TMS group was removed using TBAF, and the double bond was oxidatively cleaved with OsO_4_ and *N*-methylmorpholine-*N*-oxide (NMO) followed by treatment with NaIO_4_ delivering aldehyde **115**. For the generation of the last stereogenic center, the authors chose a stereoselective vinylogous Mukaiyama aldol reaction between aldehyde **115** and **116** in the presence of BF_3_∙OEt_2_ obtaining the dihydroxy ester **117** in a yield of 80% in the form of a single diastereomer. For the synthesis of **111a**, the absolute configuration of C5 was reversed by oxidizing the secondary alcohol using Dess–Martin periodinane and reducing the resulting ketone with NaBH_4_ stereoselectively. Then, the ester was hydrolyzed and the free alcohol was protected as a TMS ether, giving **111a**. The same two-step sequence afforded **111b**.

On the other hand, for the synthesis of carboxylic acids **111c** and **111d**, a *syn*-Evans aldol reaction between 5-hexynal and the boron enolate of the chiral oxazolidinone **118** was affected, leading quantitatively to the *syn*-aldol product **119** as a sole stereoisomer. Transamination, silylation of the hydroxyl group with TBSOTf, and reduction of the resulting Weinreb amide with DiBAL-H delivered aldehyde **120**. The same sequence of reactions described for obtaining **111a** and **111b** from aldehyde **115** was employed to produce carboxylic acids **111c** and **111d** from aldehyde **120** via the C5 epimers **121a** and **121b**.

As shown in Scheme 18, subunit **122**, composed of a tetrapeptide fragment and a hydroxy ester moiety, was synthesized in a stepwise method starting from Boc-sarcosine benzyl ester in 64% overall yield.

Having subunit **122** in hand, the next step was its condensation with carboxylic acids **111a**, **111b**, **111c**, and **111d** using EDCI∙HCl as a coupling agent (Scheme 19). C5 selective desilylation delivered alcohols **123a**, **123b**, **123c**, and **123d**, respectively, which were coupled with Fmoc-*N*-Me-l-Ala by reaction with 2,4,6-trichlorobenzoyl chloride in the presence of Et_3_N and DMAP. Cleavage of the 2,2,2-trichloroethyl and Fmoc protecting groups followed by the macrolactamization and desilylation afforded palau’amide (**5**) and three stereoisomers **124**, **125**, and **126**.

### 4.4. Odoamide *(**6**)*

The first synthetic studies concerning odoamide (**6**) and its analogues were reported by Sueyoshi et al. in 2016 [6]. First, they elucidated the structure of **6** using 1D and 2D NMR analyses as well as chemical degradation followed by chiral HPLC analysis. Moreover, they synthesized **127a**, **127b**, **127c**, and **127d**, four stereoisomers of the polyketide moiety (Scheme 20). **127a** and **127b** were converted into triols **128a** and **128b** to compare them with the triol obtained by the reduction of **6** with LiAlH_4_ and determine the absolute configuration at C8. The ^1^H NMR spectrum of synthetic **128a** matched with the one of triol obtained from the natural product **6**, revealing that the absolute stereochemistry of the stereocenters of the polyketide segment was 5*S*,6*S*,7*R*,8*S*.

For the synthesis of **127c**, the boron enolate of *N*-acylated oxazolidinone *ent-***118** was reacted with aldehyde **129** to provide the corresponding *syn*-aldol product **130**. Protection of the free alcohol gave **131** and reductive removal of the chiral auxiliary conducted to **132**. Swern oxidation to the corresponding aldehyde followed by a Wittig reaction with ethyltriphenylphosphonium bromide using *n*-BuLi as a base led to alkene **133a** in 72% from **132** as a mixture of two stereoisomers. Hydrogenation using Pd/C as a catalyst reduced the double bond with the concomitant removal of the benzyl group yielding **134a** in 85% yield. Swern oxidation of the deprotected primary alcohol **134a** and coupling with **135** in the presence of BF_3_OEt_2_ afforded the desired dihydroxy ester **127c** via a vinylogous Mukaiyama aldol reaction. **127a** was obtained starting from aldehyde *ent***-129**, which was transformed into **136a** following the same sequence as for **127c**. The synthesis of **127a** required the inversion of the newly generated stereogenic center in **136a** by oxidizing the hydroxyl group using Dess–Martin periodinane and reducing stereoselectively the corresponding ketone with NaBH_4_ in 83% yield after two steps and a dr > 97:3. The four-step sequence involving the TBS cleavage, diol protection as acetonide to give **137a**, ester reduction with DiBAL-H (**138a**), and acetonide removal gave rise to the triol **128a**.

For the synthesis of **127d**, aldehyde **129** was also used as starting material. Thus, **129** was reacted with the boron enolate of the chiral oxazolidine **139**, giving rise to the *syn*-aldol product **140**. Protection of the hydroxyl group as TBS ether gave **141** and reductive removal of the chiral auxiliary led to adduct **142**, in which the alcohol was tosylated and reduced with LiAlH_4_, giving rise to compound **133b**. As depicted in Scheme 20, the synthesis of **127d** and **127b** and the transformation of the latter into triol **128b** was achieved analogously as for **127c**, **127a**, and **128a**, respectively.

As continuation of their research in this field [28], this Japanese team completed the first total synthesis of odoamide (**6**) in the same year (Scheme 21). With the polyketide fragment **127a** in hand, the next step was incorporation of the α-hydroxy acid and polypeptide units. This was accomplished after protection of the free alcohol and saponification of **143**, by coupling d-*allo*-isoleucic acid ester **144** to the all-protected subunit **145**. Then, the latter was treated with HF pyridine to remove the TBS group, giving subunit **146**. Ester formation by reaction with Fmoc-*N*-Me-Ala-Cl using DIPEA as a base followed by Fmoc group cleavage furnished **147**. The free *N*-methylamine was used to couple this adduct with tetrapeptide **148** using EDCI–HOAt, giving rise to **149** as a 1.4:1 mixture of two epimers at C5′. After the cleavage of phenacyl and Fmoc groups with Zn/AcOH and Et_2_NH, respectively, the resulting mixture of epimers could be separated, and the desired major one **150a** was transformed into odoamide (**6**) by a macrolactamization employing HATU and HOAt and removal of the MTM group with AgNO_3_ and 2,6-lutidine. NMR spectra of synthetic odoamide (**6**) showed to be identical to those of the natural product. The minor stereoisomer **150b** was converted by the same sequence into the epimer of odoamide at C5′ **151**. Then, the cytotoxicity of **6** and **151** against A549 cells was evaluated. Whereas depsipeptide **6** showed to be highly cytotoxic (IC_50_ = 2.1 nM), **151** showed less potent antiproliferative activity (IC_50_ = 0.54 μM), demonstrating that the absolute configuration at C5′ is essential for the cytotoxicity activity of odoamide (**6**).

Inspired by these results, the authors undertook an SAR study involving modifications in the tetrapeptide domain [10]. Thus, as shown in Scheme 22, a novel odoamide (**6**) synthesis using a solid-phase peptide synthesis was reported. For this purpose, dihydroxy ester **146** was acylated with Fmoc-*N*-Me-l-Ala-Cl, the phenacyl ester was deprotected, and the resulting carboxylic acid loaded onto Cl-(2-Cl)Trt resin provided, after Fmoc group removal, compound **152**. Then, Alloc-Ile-OH was coupled, and the Alloc protecting group was cleaved by reaction with Pd(PPh_3_)_4_ and PhSiH_3_. The other four amino acids of the peptide segment were incorporated by sequential coupling of their Fmoc derivatives. Cleavage of the pentapeptide from the resin afforded amino acid **153**. Macrolactamization with HATU and HOAt afforded odoamide (**6**).

The same synthetic route was employed for the preparation of a series of 11 analogues of the natural product (Figure 5). The authors studied the intramolecular ester exchange between C5 and C7 alcohols at the polyketide moiety in odoamide (**6**) under aqueous conditions and concluded that this depsipeptide is in slow equilibrium between 26-membered macrocycle **6** and 24-membered form **154**. However, the 24-membered cyclic **154** exhibited highly potent cytotoxicity against A549 cells, despite the apparently alternative global conformations (IC_50_ = 4.5 nM for **154** and IC_50_ = 4.2 nM for **6**). The C7-methoxy derivative **155**, which cannot undergo 1,3-acyl transfer reaction, exhibited low activity, while C5-methoxy derivative **156** retained cytotoxicity. The conjugated diene **157**, lacking the C5 hydroxyl group, the *N*-methyl-β-alanine, and the glycine analogues **158** and **159** showed slightly lower cytotoxicity. The absolute configurations at C5′ and C14′ showed to be crucial, since C5′ and C14′-epimers **160** and **161** were much less active (IC_50_ = 1554 nM) than the parent compound **6**. The inversion of the C11′ configuration resulted in a 10-fold decrease in the toxicity (**162**: IC_50_ = 48 nM), but the loss of the *N*-methyl group of C11′ was tolerated (**163**: IC_50_ = 1554 nM). Finally, the C2′ epimer **164**, bearing a *N*-Me-d-Ala showed more potent cytotoxicity than natural odoamide.

### 4.5. Lagunamides A *(**7**)* and B *(**8**)*

In 2010, Tripathi and co-workers reported the isolation and the planar structural characterization of lagunamides A and B [7] (Figure 6).

Two years later, Dai et al. proposed a revised configuration assignment for lagunamide A (**7**), which was validated by total synthesis [29].

Thus, initial investigations focused on the preparation of the supposed structure **165** (Scheme 23). For this, tetrapeptide **167** was obtained by coupling dipeptides **168** and **169** using HATU as the coupling reagent, delivering *N*-Boc protected tetrapeptide **167** after saponification of the methyl ester. Phosphonate **170** was accessed by hydrolysis and condensation, using *N*,*N*-diisopropylcarbodiimide (DIPC) as the coupling agent, with 2-(diethyoxyphosphoryl)propanoic acid from diester **171**, which was itself synthesized from commercial available D-*allo*-isoleucine.

Next, they investigated the synthesis of aldehyde **172a** and coupling with phosphonate **170** via Horner–Wadsworth–Emmons olefination (Scheme 24). Thus, the stereocenters C6 and C7 were installed by reaction of the boron enolate of chiral oxazolidinone *ent-***118** with *S*-2-methylbutanal, affording a *syn*-aldol product, which was transformed into acetal **173** by reduction with NaBH_4_ followed by reaction of the corresponding diol with anisaldehyde dimethyl acetal in the presence of PPTS. Reductive cleavage of the anisylidine acetal with DiBAL-H, Dess–Martin oxidation of the resulting primary alcohol in aldehyde, and treatment in situ with allyltributylstannane in the presence of BF_3_·OEt_2_ delivered the homoallylic alcohol **174a** (67%) along with its diastereomer **174b** (27%). The free hydroxyl group was protected as its triethylsily (TES) ether, and the oxidative cleavage of the double bond afforded aldehyde **172**. Horner–Wadsworth–Emmonds condensation between phosphonate **170** and aldehyde **172** led to the α,β-unsaturated ester **175**. Reaction with DDQ to remove the PMB protecting group gave a partial deprotection of the TES group delivering a mixture of free diol **176** (32%) and the monoprotected diol **177** (64%). Consequently, a selective re-protection of the most accessible hydroxyl was performed. Coupling of **177** with Fmoc-*N*-Me-Ala-Cl in the presence of DMAP led to triester **178**.

Having fragments **167** and **178** in hand for their assembly, deprotection of the Fmoc group in **178** was followed by coupling using HATU and HOAT, which furnished the *N*-Boc-protected precursor **179** (Scheme 25). The simultaneous cleavage of TES, Boc, and *t*Bu groups with TFA followed by macrolactamization delivered **165**.

Following the same sequence, homoallylic alcohol **174b** was used as a precursor for the synthesis of diastereoisomer of lagunamide A **180**. Neither ^1^H nor ^13^C spectra of both **165** and **180** corresponded to those of the natural product.

To confirm the assignment of the stereochemistry of **165**, the authors developed a second synthetic pathway to access **176** (Scheme 26). After protection of the free hydroxyl group as its TES ether in **181** followed by oxidative cleavage of the double bond, the corresponding aldehyde was transformed into homoallylic alcohol **182** via an *anti* crotylation using the chiral organoborane reagent (−)-Ipc_2_-BOMe with BF_3_OEt_2_ as a mediator. TES protecting group cleavage followed by reaction with DMP in the presence of PPTS afforded acetonide **183**. Primary alcohol deprotection and oxidation with Dess–Martin periodinane gave rise to aldehyde **184**, which was reacted with phosphonate **170** obtaining, via Horner–Wadsworth–Emmonds condensation, compound **176**, after diol deprotection. Compound **176** proved to be identical to the derivative obtained by following the synthetic pathway described in Scheme 24.

Then, looking for the actual structure of lagunamide A, the authors focused on the polyketide moiety and synthesized epimers **185** and **186**. For this purpose, they prepared in three steps the diprotected triols **187** and **188** starting from **189**, as outlined in Scheme 27, which were introduced in macrocycles **185** and **186**. Both were found to be different from the natural lagunamide A (**7**).

Finally, they focused on modifying the tetrapeptide fragment by preparing unnatural analogues **190** and **191**, which included the polyketide moiety of **186** and **185** respectively but in which l-*allo*-isoleucine was replaced with L-isoleucine (Figure 7). To their delight, the ^13^C NMR data, HRMS, and specific optical rotation of synthetic **191** matched those of the natural product lagunamide A (**7**). This outstanding work proposed and validated a revised configurational assignment by total synthesis for lagunamide A and provided three non-natural analogues.

In 2013, Huang and co-workers presented an asymmetric total synthesis of lagunamide A as well as five non-natural analogues [30]. For the synthesis of the polyketide fragment (Scheme 28), the key step was the *syn*-Evans aldolisation between the boron enolate of (*R*)-4-benzyl-3-propionyloxazolidin-2-one *ent*-**118** and aldehyde **192**, obtained from commercially available (*S*)-2-methylbutan-1-ol, which delivered the *syn*-aldol **193**, in which the three of the four stereocenters of the polyketide moiety were installed. Then, the hydroxyl group was protected as TBS ether, and the chiral auxiliary was cleaved by reduction upon treatment with LiBH_4_, leading to **194**. Then, the primary alcohol was oxidized following Swern’s protocol, and the allylation of the subsequent aldehyde **195** with allylMgCl, in the absence of Lewis acid, produced a mixture of homoallylic alcohols **196a** (11%) and **196b** (81%). By employing zinc chloride, the selectivity could be reversed, giving **196a** and **196b** in a 90:10 diastereomeric ratio.

Having compound **196a** in hand, sequential protection of the free alcohol as a Troc and removal of the TBS ether led to **197** (Scheme 29). Reaction with Fmoc-*N*-Me-l-Ala-Cl using DIPEA as a base furnished amino ester **198** in which the *N*-Fmoc protection was replaced by *N*-Boc to give **199** in 75% overall yield. Cross-metathesis of terminal olefin with tiglic aldehyde using Grubbs II catalyst yielded α,β-unsaturated aldehyde **200**, which in turn was converted into carboxylic acid **201** by a Pinnick oxidation. The same three-step sequence allowed the preparation of carboxylic acid **202** from **203** via aldehyde **204**.

The synthesis of the peptide fragment is shown in Scheme 30. Removal of the Boc group in sarcosine derivative **205** followed by condensation with Boc-*N*-Me-d-Phe-OH in the presence of HATU/DIPEA delivered the protected amino acid **206** in 89% yield from **205**. The same two-step sequence afforded tripeptide **207** in 68% yield. Boc cleavage and coupling with (2*R*,3*S*)-2-hydroxy-3-methylpentanoic acid in the presence of EDC/HOBt led to **208** in 56% yield.

Having polyketide moieties **201** and **202** and peptide fragment **208** in hand, the authors turned their attention to the coupling of these units (Scheme 31). Thus, condensation of **201** with **208** was achieved using MNBA/DMAP as coupling agent affording **209** in 66% yield. Boc cleavage and coupling with Boc-l-*allo*Leu-OH using HATU and DIPEA produced **210** in 82% yield. Allyl and Boc deprotection upon treatment with Pd(PPh_3_)_4_ and TFA respectively followed by macrolactamization with HATU/DIPEA and concomitant elimination of Troc protecting group led to **211** in 22% yield from **210**. As shown in Scheme 31, the same sequence starting from **202** and **208** was conducted to the desired analogue **214** via compounds **212** and **213**. Starting from **212**, by allyl deprotection and coupling with l-lle-*O*-allyl, employing HATU/DIPEA, **215** was obtained in 85% yield. The synthesis of analogue **216** was completed using a series of four transformations. Sequential removal of the allyl and Fmoc groups paved the way for macrolactamization using HATU. Cleavage of the TBS group resulted in the desired lagunamide A analogue.

Next, the authors focused on the preparation of the new lagunamide A analogue **217** starting from **196b** (Scheme 32). For synthetic reasons, the TBS ether was permuted by deprotecting the C6 hydroxyl group with aqueous HF and protection of the most accessible C4 one using TBSOTf and 2,6-lutidine at low temperature in 92 and 75% yield, respectively. The free alcohol of the resulting product **218** was used for a coupling with Fmoc-*N*-Me-l-Ala-Cl. A cross-metathesis of terminal olefin with tiglic aldehyde catalyzed by Grubbs II catalyst afforded aldehyde **219**. After a Pinnick oxidation, the corresponding carboxylic acid **220** was condensed with the peptidic fragment **208** using MNBA as the coupling agent, leading to **221** in 63% yield. The sequence Fmoc deprotection and coupling with Boc-l-*allo*Leu-OH conducted to **222**. Boc and TBS groups removal and macrolactamization afforded the lagunamide A analogue **217**.

To conclude, the synthesis of the natural product lagunamide A (**7**) and its synthetic analogue 2-*epi*-lagunamide A (**223**) was carried out (Scheme 33). The ester **224** was first subjected to a cross-metathesis of terminal olefin with tiglic aldehyde employing Grubbs II catalyst to afford aldehyde **225**. Pinnick oxidation and coupling of the corresponding carboxylic acid **226** to the peptide fragment **208** conducted to **227** in 61% yield. The sequence allyl group cleavage, and coupling with Boc-l-*allo*Ile-OH produced **228**, which after allyl and Fmoc deprotection, macrolactamization, and TBS removal afforded lagunamide A (**7**). The same sequence of reactions starting from **229** led to 2-*epi*-lagunamide A (**223**).

The stereoselective synthesis of fragment C27–C45 of lagunamide A **233** was reported by Chang’s group in 2014 (Scheme 34) [31]. Dimethyl acetal **234**, prepared from commercially available (*S*)-2-methylbutanal, served as the point of departure of the synthesis. Thus, the acetal aldol reaction between **234** and thiazolidinethione **235** provided the *anti*-methylated aldol product **236** in a diastereomeric ratio of 82:18 and 62% yield. The chiral auxiliary was removed with DiBAL-H, and the resulting aldehyde was reacted with allylmagnesium chloride, giving rise to the desired derivative **237a** as well as its diastereomer **237b** in 99% yield in a ratio of 2:3. Consequently, the absolute configuration at the C5 had to be reversed in a two-step sequence involving Dess–Martin periodinane oxidation of the free alcohol in **237b** followed by the diastereoselective reduction of the corresponding ketone **238** with NaBH_4_, which delivered the homoallylic alcohol **237a** in 95% yield and a ratio of 8:1. The newly generated hydroxyl group was masked as its TBS ether, and the resulting adduct **239** was subjected to a cross-metathesis with methyl methacrylate using Grubbs II catalyst affording **240**. Saponification of the ester using LiOH released the corresponding free carboxylic acid **241**, which was coupled to the α-hydroxy ester **242** using DCC and DMAP, affording the desired aliphatic segment **233**. The synthetic utility of this approach is hampered by the lack of stereoselectivity in the generation of homoallylic alcohol 237 and the choice of a non-removal methylether in C7.

In 2016, Banasik et al. published an elegant synthesis of fragment C27–C45 **243** of lagunamide A (**7**) involving two iterative vinylogous Mukaiyama aldol reactions (VMAR) (Scheme 35) [32]. Thus, chiral vinylketene silyl *N*,*O*-acetal **244** was reacted with the commercially available (*S*)-2-methylbutanal, leading to the corresponding *anti*-aldol product **245** in 96% yield and an excellent diastereomeric ratio (>98:2). Then, the free alcohol was protected as a propionate ester (**246**) and ozonolysis was conducted to aldehyde **247**, which in turn was involved in the second VMAR with vinylketene silyl *N*,*O*-acetal **248** giving rise to *anti*-aldol **249**. The synthesis was achieved by the protection of the hydroxyl group as a BOM ether and oxidative removal of the chiral auxiliary, which delivered the free carboxylic acid **250**. The latter was esterified with the α-hydroxy ester **251** using DCC and DMAP affording the protected C27–C45 fragment **243** of lagunamide A (**7**). This original synthetic route included unexplored asymmetric transformations but was penalized by the low yield of the second VMAR.

Despite the progress made in the synthesis of compounds of the aurilide class, the development of alternative strategies remains an important research goal, in particular with regard to the generation of the stereocenters in the aliphatic moiety. In this context, Gorges et al. have recently reported a new strategy for the total synthesis of lagunamide A (**7**) based on the stereoselective homologation of boronic esters of C_2_-symmetrical chiral diols developed by Matteson [33]. This chiral auxiliary-based methodology was applied to achieve control of the absolute stereochemistry in the synthesis of the polyketide fragment of lagunamide A (**7**).

As depicted in Scheme 36, the synthesis commenced with chiral boronic ester **252**, which was involved in a first Matteson homologation using ethylmagnesium bromide as nucleophile affording **254,** via the intermediate **253**, in excellent yield and diastereoselectivity. Three more successive homologations employing NaOBn, MeMgCl, and NaOPMB as nucleophiles afforded the prolonged boronic esters **255**, **256**, and **257** respectively. Having generated all the stereocenters of the polyketide moiety, a further homologation aimed at the introduction of a methylene and was succeeded by reaction with dibromomethane and *n*-BuLi at −60 °C, giving rise to **258**. The last homologation of **258** afforded the (α-chloroalkyl)boronic ester **259**, which was subjected to oxidation in aldehyde **260**. Protection of the aldehyde as an acetal and benzyl removal yielded the monoprotected diol **261**, which, after coupling to Fmoc-*N*-Me-l-Ala-Cl and acetal hydrolysis, afforded aldehyde **262**.

As illustrated in Scheme 37, the authors envisaged coupling polyketide precursor **262** to the α-hydroxy acid subunit by a Horner–Wadsworth–Emmons reaction between aldehyde **262** and phosphonate **170**. To this end, **170**, prepared from the chiral allylic ester **263**, gave, via Ireland–Claisen rearrangement, the protected α-hydroxy ester derivative **264** in 73% yield. The carbon chain of this compound was shorted by a metathesis using Grubbs II catalysts under an ethylene atmosphere delivering **265** in 77% yield. A three-step sequence involving the reduction of the double bond and simultaneous debenzylation followed by the coupling of the free alcohol with 2-(diethyoxyphosphoryl)propanoic acid conducted to the expected phosphonate **170** in 90% yield. Having building blocks **262** and **170** in hand, Horner–Wadsworth–Emmons reaction could be implemented, and **266** could be accessed using the lithium salt of hexafluoroisopropanol (HFIP) as a base in 65% yield (Scheme 37). A partial epimerization of the *N*-methylalanine was observed. Then, the Fmoc protecting group was cleaved, and the resulting free secondary amine was coupled with Fmoc-isoleucine to afford **267** in 91% yield.

In order to complete the pentapeptide fragment, the requisite protected tripeptide **268** was previously prepared starting from protected sarcosine **269**, which was successively coupled with Boc-*N*-Me-d-Phe-OH and Boc-l-Ala-OH using EDC and HOBt or HOAt in the presence of DIPEA, and the ester was saponified, leading to **268** (Scheme 38). After removal of the Fmoc protecting group in **267** and coupling with tripeptide **268** using 1-[(1-(cyano-2-ethoxy-2-oxoethylideneamino-oxy)-dimethylamino-morpholinomethylene)] methanaminium hexafluorophosphate (COMU), **271** was obtained. After a simultaneous removal of Boc, *tert*-butyl, and PMB protecting groups using trifluoroacetic acid, the ring closing was achieved using HATU and HOAt under high dilution, affording lagunamide A (**7**). The authors have shown, with this arduous but meaningful path, that the Matteson homologation is an excellent tool for the synthesis of the polyketide moiety of lagunamide A that they prepared via six iterative Matteson homologation steps and subsequent oxidation with an overall yield of 30%. Indeed, this original approach offers high stereoselectivities using different nucleophiles. For the synthesis of subunit 170, they developed an innovative strategy. Unfortunately, the tetrapeptide fragment had to be introduced stepwise to avoid the epimerization of the C-terminal isoleucine in **266**.

In the field of their synthetic studies concerning lagunamide B (**8**), Pal et al. published the stereoselective synthesis of an advanced analogue of this depsipeptide [34]. As shown in Scheme 39, the synthesis began with Crimmins *syn* aldolisation between Ti enolate of acylated chiral oxazolidinone *ent***-118** and tiglic aldehyde, giving rise to the *syn* aldol product **272** in 88% yield and with a diastereomeric ratio of 97:3. Then, the free alcohol was protected as TBS ether. Reductive removal of the chiral auxiliary followed by oxidation of the resulting primary alcohol delivered aldehyde **273**, which was subjected to reaction with boron enolate of the chiral oxazolidinone **274**. Two diastereomers **275a** and **275b** were obtained in 63% yield and a ratio of 2:1. Then, major steroisomer **275a** was protected as MOM-ether and reductive cleavage of the chiral auxiliary conducted to a primary alcohol, which was oxidized using pyridine–sulfur trioxide complex and Et_3_N conducting to aldehyde **276**. The α-hydroxy ester **277**, obtained from l-Ile, was condensed with 2-(diethoxyphosphoryl)propanoic acid using DIC as a coupling agent, and the resulting adduct was coupled with aldehyde **276** via Horner–Wadwords–Emmonds reaction, leading to fragment **278** in 59% yield, after benzyl removal by hydrogenation.

The next phase of the synthesis was the preparation of the tripeptide fragment **279** (Scheme 40). Thus, Boc-*N*-Me-D-Phe-OH and HCl∙H-Gly-OMe were coupled using EDCl and HOBt. The resulting dipeptide was submitted to a four-step sequence involving the *N*-methylation with methyl iodide and silver oxide, Boc deprotection with TFA, coupling with Boc-Ala-OH using HATU and DIPEA as a base, and saponification of the methyl ester to give the *N*-Boc-protected tripeptide **279**. Then, the amine of the amino acid derivative Boc-Ala-OMe was methylated, the Boc protecting group was removed, and the resulting *N*-methyl ester was coupled with Boc-Ile-OH using HATU/DIPEA to furnish dipeptide **280** after Boc deprotection using TFA.

Tetrapeptide **281** was finally obtained by coupling units **279** and **280** by using the system EDCl/HOBt/DIPEA in 72% yield followed by methyl ester saponification, benzylation of the resulting free acid, and Boc removal.

Having units **278** and **281** in hand, they were coupled using EDCl and HOBt with DIPEA as a base, and the TBS ether was cleaved, affording **282**. The synthesis was completed by debenzylation with PdCl_2_, Et_3_N, and Et_3_SiH and macrolactonization using MNBA and DIPEA delivering the MOM-hydroxy protected lagunamide B analogue **283**, which was obtained in a low yield.

Elucidation of the molecular target of lagunamide A (**7**) could be essential in order to determine the mechanism of action of this natural product. In this context, Banasik has recently disclosed the synthesis of a biotin-linker moiety **284**, with the aim to be coupled to lagunamide A (**7**) by the C5 hydroxyl group (Figure 8) [35].

### 4.6. Lagunamides C *(**9**)*

Lagunamide C (**11**) is the last member of the aurilide-class cytotoxic cyclic depsipeptides, which was isolated by Tripathi and co-workers from the cyanobactium *Lyngbya majuscule* [8]. These authors also reported the complete structural characterization.

Fatino et al. recently explored a new route to access the polyketide moiety of lagunamide C (**9**) (Scheme 41) [36]. The challenge posed by this fragment (**285**) is the insertion of the extra methylene carbon at C7. The retrosynthetic approach involved the preparation of intermediate **286** via a route, which had as its key step a cyclopropanation and subsequent ring opening. The authors decided to develop this process using butanal, devoid of the C9 methyl group, as the starting aldehyde. This aldehyde was reacted with acetylated thiazolidinethione **287** using LDA as a base in the presence of TiCl_4_ and *N*-methylpyrrolidone (NMP). The subsequent hydroxyamide **288** was transformed into aldehyde **289** by a three-sequence step involving the protection of alcohol as TBS ether, NaBH_4_ reduction, and oxidation under Swern conditions of the primary alcohol. Reaction of **289** with the activated ylide methyl(triphenylphosphoranylidene)acetate provided the allylic methyl ester **290** in 80% yield, which was reduced to the primary alcohol **291** with DiBAL-H. Cyclopropanation with Et_2_Zn, CH_2_I_2_ at 0 °C yielded cyclopropane **292** as a mixture of diastereomers. Mesylation of the primary alcohol followed by reaction with NaI afforded iodide **293**, which was subjected to *n-*BuLi and tetramethylethylene diamine (TMEDA), delivering a mixture of terminal alkenes **295a** and **295b** in 71% combined yield. The ratio of **295a** and **295b**, which both lack the C9 methyl group of natural lagunamide C (**9**), was not reported by the authors.

## 5. Conclusions

In conclusion, aurilide and other members of this family of marine natural products are potent cytotoxic agents that constitute a serious option for the design of new anticancer drugs. These compounds induce apoptosis in various human cell lines at the picomolar to nanomolar range. Biochemical and molecular biological investigations are focusing their efforts on identifying the target proteins responsible for the observed apoptosis. Some leads are already the subjects of intensive research, such as prohibitin 1 (PHB1), whose function is inhibited by binding with aurilide, resulting in mitochondria-induced apoptosis.

Lagunamides A and B cytotoxicity has also been reported to be related to mitochondrial mediated apoptosis through the overproduction of reactive oxygen species (ROS).

Despite their complex structure, several total syntheses gathered in this review proved to be efficient to produce natural compounds of aurilide class, but also several analogues have been synthesized. Their evaluation against cancer cells enables studying structure–activity relationships and dictating certain rules for categorizing what is determinant in structure for anticancer activity.

All these data underline the importance of the aurilide class of molecules to serve as a chemical tool to investigate mitochondrial-mediated regulation of apoptosis and encourage more in-depth explorations of potential anti-cancer drugs.

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
