# Peer review of "Synthesis and Biological Activities of Cyclodepsipeptides of Aurilide Family from Marine Origin"

_marinedrugs, 2021, doi:10.3390/md19020055_

Round 1

Reviewer 1 Report

Title: Synthesis and Biological Activities of Aurilide Family Cyclodepsipeptides from Marine Origin

Article type: Review

This review article summarises aurilides, a class of depsipeptides found in marine cyanobacteria and the total synthesis of naturally occurring aurilides as well as some synthetic analogues. Authors have taken a substantial effort to summarise the literature related to isolation and characterisation of naturally occurring aurilides, the cytotoxic activity of the class and the synthesis of aurilides. Unfortunately, authors have missed the main point of a review article, a comprehensive analysis of literature. Particularly, in the section 4, authors have re-written the methodology section of the corresponding literature and have reproduced the synthetic schemes described in the original article. This has resulted in a very lengthy review article with poor critical analysis of literature. It would be very interesting to see a critical analysis of the different synthetic approaches used to synthesise various aurilides analogues. 

Overall, this is a poorly written review article. There are plenty of technical, formatting and grammatical errors throughout the manuscript. Authors have paid minimal attention to complete this as a well-written review article. I have a few other comments on this work other than the above mentioned significant issues, and authors should address all these to make this work publishable.

Comment 01: Incorrect in-text citation and bibliography.

Page 1, line 23: Yamada is the corresponding author of this work, not the first author (reference 2). Authors should use the first author "Suenaga" for the in-text citation. This is a common error throughout the manuscript.

In line 224, Kigoshi and co-workers [17] should be corrected as "Takahashi et al. or Takahashi and co-workers.

In line 289, J. Kimura and co-workers should be corrected as "Nakaoet al. or Nakao and co-workers.

Moreover, there are many formatting errors in the bibliography.

Comment 02: A poor language has been used throughout the manuscript.

Following are a few examples only;

Line 30, should be corrected as "source of aurilide B"

Line 53, should be correct as "can be found"

Line 63, should be corrected as "unsaturated"

Line 92, should be corrected as "an inducer" and "morphogenesis"

Line 276, "is hardly less cytotoxic" ?

Comment 03: Authors have used non-standard terms.

For example, line 292, "K. Yamada's method" Is this a well-established synthetic methodology or a method described in Yamada's work?

This type of language can be found throughout the manuscript.

Author Response

Object:    Marine Drugs - Manuscript ID: 1067459

Title: Synthesis and Biological Activities of Cyclodepsipeptides of Aurilide Family from Marine Origin

Dear Editor,

We were very pleased to read that the reviewers found this manuscript to reflect a valuable scientific contribution for publication in your special issue of Marine Drugs. The reviewers also raised several points of discussion requiring additional explanation, which we are happy to provide.

You will thus find enclosed a revised version of the manuscript in which the changes are highlighted in yellow. Some schemes have also been corrected: 2, 3, 6, 12, 16, 17, 25, 31, 35. We have also explained point-by-point how we have addressed the reviewer’s suggestions.

The graphical abstract has been changed and a picture without any text is now proposed.

We thank the reviewers for their important and constructive comments and feel that their review has enabled us to improve the manuscript significantly. We hope that the provided explanations and revision of our manuscript will be found suitable for publication.

Sincerely yours,                                                                      

                                                           Florine Cavelier and Xavier J. Salom-Roig

Reviewer 1

This review article summarises aurilides, a class of depsipeptides found in marine cyanobacteria and the total synthesis of naturally occurring aurilides as well as some synthetic analogues. Authors have taken a substantial effort to summarise the literature related to isolation and characterisation of naturally occurring aurilides, the cytotoxic activity of the class and the synthesis of aurilides. Unfortunately, authors have missed the main point of a review article, a comprehensive analysis of literature. Particularly, in the section 4, authors have re-written the methodology section of the corresponding literature and have reproduced the synthetic schemes described in the original article. This has resulted in a very lengthy review article with poor critical analysis of literature. It would be very interesting to see a critical analysis of the different synthetic approaches used to synthesise various aurilides analogues. 

The authors thank the reviewer for this very comprehensive and concise summary.

As indicated by the reviewer, the main ambition of this review is to gather in a single article all the literature reporting synthesis and biological evaluation of aurilides.

The reviewer is however a little severe in his remarks, the review certainly reports data from the literature, this is its primary objective, but also comments on them, with research on the attributions of absolute configurations, the key points of the structure-relationship activity, the interest of analogues... One cannot say that it is about an assembly of data of the literature without construction. For each compound, we narrated a story, with its chronology, from the discovery of the natural product to the identification of its structure, the confirmation of the absolute configurations, the syntheses by different strategies, and the synthesis of analogues if there are some, and why they have been synthesized, with an effort of clarity and conciseness. However, following the suggestion of the reviewer of adding more critical analysis, we have inserted some supplementary personal comments (highlighted in yellow).

Overall, this is a poorly written review article. There are plenty of technical, formatting and grammatical errors throughout the manuscript. Authors have paid minimal attention to complete this as a well-written review article.

The manuscript has been carefully proofread and many typos and grammatical errors have been corrected, including some errors in schemes.

I have a few other comments on this work other than the above mentioned significant issues, and authors should address all these to make this work publishable.

Comment 01: Incorrect in-text citation and bibliography.

Page 1, line 23: Yamada is the corresponding author of this work, not the first author (reference 2). Authors should use the first author "Suenaga" for the in-text citation. This is a common error throughout the manuscript.

In line 224, Kigoshi and co-workers [17] should be corrected as "Takahashi et al. or Takahashi and co-workers.

In line 289, J. Kimura and co-workers should be corrected as "Nakaoet al. or Nakao and co-workers.

Everywhere, the first author is now cited with the reference.

Moreover, there are many formatting errors in the bibliography.

As requested, all the references have been formatted.

Comment 02: A poor language has been used throughout the manuscript.

Following are a few examples only;

Line 30, should be corrected as "source of aurilide B"

Line 53, should be correct as "can be found"

Line 63, should be corrected as "unsaturated"

Line 92, should be corrected as "an inducer" and "morphogenesis"

Line 276, "is hardly less cytotoxic" ?

The suggested changes have been made as well as a careful revision of the whole manuscript.

Comment 03: Authors have used non-standard terms.

For example, line 292, "K. Yamada's method" Is this a well-established synthetic methodology or a method described in Yamada's work?

This type of language can be found throughout the manuscript.

We understand what the reviewer means and we have adapted the text accordingly, removing the apostrophe of possession.

Reviewer 2 Report

Authors describe a nice review of the syntheses of aurilide and its derivatives and their potent biological activities. It will make a good impression on the readers. Publication is recommended after some revisions. There are many errors, which are pointed in the pdf file attached.

Author Response

Object:    Marine Drugs - Manuscript ID: 1067459

Title: Synthesis and Biological Activities of Cyclodepsipeptides of Aurilide Family from Marine Origin

Dear Editor,

We were very pleased to read that the reviewers found this manuscript to reflect a valuable scientific contribution for publication in your special issue of Marine Drugs. The reviewers also raised several points of discussion requiring additional explanation, which we are happy to provide.

You will thus find enclosed a revised version of the manuscript in which the changes are highlighted in yellow. Some schemes have also been corrected: 2, 3, 6, 12, 16, 17, 25, 31, 35. We have also explained point-by-point how we have addressed the reviewer’s suggestions.

The graphical abstract has been changed and a picture without any text is now proposed.

We thank the reviewers for their important and constructive comments and feel that their review has enabled us to improve the manuscript significantly. We hope that the provided explanations and revision of our manuscript will be found suitable for publication.

Sincerely yours,                                                                    

                                                       Florine Cavelier and Xavier J. Salom-Roig

Reviewer 2

We would like to thank the reviewer for sending us an in-text corrected manuscript that was very useful to help in correcting all errors to highly improve the quality of our review. We have the corrections pointed out. In particular, we have detailed the signification of some abbreviations used in the text and schemes at the end of the corresponding scheme captions (highlighted in yellow):

-DEPC and EDCI, scheme 2

-DMAP, scheme 3

-DIEA, scheme 4

-PPTS, scheme 11

-PMBTCA, scheme 13

-MNBA, scheme 31

Moreover, the abbreviation NMA, in scheme 15, has been replaced by PhNHMe and the abbreviation DMPQ, in scheme 37, has been replaced by DMPA (it was a typo).

Reviewer 3 Report

The article is interestingly, excellently designed and edited. I recommend publishing it in this current form. Please only check and modify were you deem it necessary:

line 92 - porhogenesis? or morphogenesis

145 - propiponyl?

198 -point after Scheme 5

407- space after (5).

420 and 433 TMS-C5H6MgBr?

436- Ph3P=CHCH3COOAllyl 

Small technical corrections to the bibliography at 998, 1008, 1018, 1040, 1047, 1060, 1062, 1064, 1067

Author Response

 Object:    Marine Drugs - Manuscript ID: 1067459

Title: Synthesis and Biological Activities of Cyclodepsipeptides of Aurilide Family from Marine Origin

Dear Editor,

We were very pleased to read that the reviewers found this manuscript to reflect a valuable scientific contribution for publication in your special issue of Marine Drugs. The reviewers also raised several points of discussion requiring additional explanation, which we are happy to provide.

You will thus find enclosed a revised version of the manuscript in which the changes are highlighted in yellow. Some schemes have also been corrected: 2, 3, 6, 12, 16, 17, 25, 31, 35. We have also explained point-by-point how we have addressed the reviewer’s suggestions.

The graphical abstract has been changed and a picture without any text is now proposed.

 We thank the reviewers for their important and constructive comments and feel that their review has enabled us to improve the manuscript significantly. We hope that the provided explanations and revision of our manuscript will be found suitable for publication.

Sincerely yours                                                               

                                                           Florine Cavelier and Xavier J. Salom-Roig

 Reviewer 3

The article is interestingly, excellently designed and edited. I recommend publishing it in this current form. Please only check and modify were you deem it necessary:

We would like to thank the reviewer for the careful reading of the manuscript and his very positive comment.

line 92 - porhogenesis? or morphogenesis

145 - propiponyl?

198 -point after Scheme 5

407- space after (5).

420 and 433 TMS-C5H6MgBr?

436- Ph3P=CHCH3COOAllyl 

These minor changes requested by the reviewer have all been addressed in the revised manuscript.

Small technical corrections to the bibliography at 998, 1008, 1018, 1040, 1047, 1060, 1062, 1064, 1067

As requested, all the references have been formatted.

Round 2

Reviewer 1 Report

Authors have addressed all the requested improvements. I would be happy to see this as a publication in the current version.